# What is the effect of changing eligibility criteria for disability benefits on employment? A systematic review and meta-analysis of evidence from OECD countries

Philip McHale[1]*, Andy Pennington[1], Cameron Mustard[2,3], Quenby Mahood[3], Ingelise Andersen[4], Natasja Koitzsch Jensen[4], Bo Burström[5], Karsten Thielen[4], Lisa Harber-Aschan[5], Ashley McAllister[5], Margaret Whitehead[1], Ben Barr[1]

1 Department of Public Health, Policy and Systems, Institute of Population Health Sciences, University of Liverpool, Liverpool, United Kingdom, 2 Dalla Lana School of Public Health, University of Toronto, Ontario, Canada, 3 Institute for Work & Health, Toronto, Ontario, Canada, 4 Department of Public Health, University of Copenhagen, Copenhagen, Denmark, 5 Department of Global Public Health, Karolinska Institutet, Stockholm, Sweden

* p.mchale@liverpool.ac.uk

## Abstract

### Background

Restrictions in the eligibility requirements for disability benefits have been introduced in many countries, on the assumption that this will increase work incentives for people with chronic illness and disabilities. Evidence to support this assumption is unclear, but there is a danger that removal of social protection without increased employment would increase the risk of poverty among disabled people. This paper presents a systematic review of the evidence on the employment effects of changes to eligibility criteria across OECD countries.

### Methods

Systematic review of all empirical studies from OECD countries from 1990 to June 2018 investigating the effect of changes in eligibility requirements and income replacement level of disability benefits on the employment of disabled people. Studies were narratively synthesised, and meta-analysis was performed using meta-regression on all separate results. The systematic review protocol was registered with the Prospective Register for Systematic Reviews (Registration code: PROSPERO 2018 CRD42018103930).

### Results

Seventeen studies met inclusion criteria from seven countries. Eight investigated an expansion of eligibility criteria and nine a restriction. There were 36 separate results included from the 17 studies. Fourteen examined an expansion of eligibility; six found significantly reduced employment, eight no significant effect and one increased employment. Twenty-two results examined a restriction in eligibility for benefits; three found significantly increased employment, 18 no significant effect and one reduced employment. Meta-regression of all studies produced a relative risk of employment of 1.06 (95% CI 0.999 to 1.014; $I^2$ 77%).

**Data Availability Statement:** All relevant data are within the manuscript and its Supporting Information files.

**Funding:** The study is part of a larger project entitled Tackling Health Inequalities and Extending Working Lives (THRIVE). The funders were The Innovation Fund Denmark (5194-00004B), the Swedish Research for Health, Working Life and Welfare (2015-01531), the UK Economic and Social Research Council (ES/N019261/1) and the Canadian Institutes of Health Research (EWL-14428). THRIVE is one of the projects of the Joint Programme Initiative More Years, Better Lives. The Innovation Fund Denmark - https://innovationsfonden.dk/en Swedish Research for Health, Working Life and Welfare - https://www.norface.net/partner/the-swedish-research-council-for-health-working-life-and-welfare-forte/ Economic and Social Research Council - https://esrc.ukri.org/ Canadian Institutes of Health Research - https://cihr-irsc.gc.ca/e/193.html The funders had no role in study design, data collection and analysis, decision to publish, or preparation of the manuscript.

**Competing interests:** The authors have declared that no competing interests exist.

## Conclusions

There was no firm evidence that changes in eligibility affected employment of disabled people. Restricting eligibility therefore has the potential to lead to a growing number of people out of employment with health problems who are not eligible for adequate social protection, increasing their risk of poverty. Policymakers and researchers need to address the lack of robust evidence for assessing the employment impact of these types of welfare reforms as well as the potential wider poverty impacts.

## Introduction

Throughout the Organisation for Economic Co-operation and Development (OECD) there is a large gap in employment between people who are disabled and those who are not [1–3]. This has an impact on health, and particularly health inequalities, as those who already have health-limiting conditions are at increased risk of job loss, social exclusion and poverty, which can further exacerbate poor health [4]. The adverse employment effects of disability are of increasing public health concern as retirement ages increase, as this will mean that an increasing proportion of the working age population (i.e. those under retirement age) will be older and will be more likely to have a disability [5]. Disability benefits aim to provide financial support for individuals who are unable to work due to chronic illness or disability, and prevent them from falling into poverty. Increasing numbers of people receiving disability benefits in many countries has led to concerns by governments that easy access to disability benefits reduces the employment of disabled people by reducing their incentives to work [2]. In response, many OECD countries have restricted access to disability benefits by narrowing eligibility criteria and introducing more stringent assessments [1].

Much of the evidence cited to support this policy response has been based on US studies which suggested that increased availability of disability benefits during the 1980's and 1990's led to a marked decline in the employment of older men in the US [6–9]. These studies however provide limited insight to inform the likely impact of current policies, particularly in the context of European countries with well-developed welfare systems and universal healthcare [10]. Firstly, the US context and the time period of these studies may not be relevant to the current situation across OECD countries. Secondly, very few of these studies investigated specific policy changes and therefore may be unable to distinguish between disincentive effects of the benefits scheme and underlying health and labour market trends. Finally, those US studies that have evaluated policy interventions have tended to investigate policies that increase access to disability benefits, while the focus of recent reforms in OECD countries is to restrict access [10]. A previous systematic review of studies up to 2009 from five OECD countries (UK, Canada, Denmark, Norway and Sweden) found there was no consistent evidence from these countries that changing eligibility criteria for disability benefits was associated with the employment of disabled people, and there was only weak evidence that higher benefit replacement rates were associated with lower employment [10]. This previous review however, had a number of limitations. Firstly, it only included studies from 5 countries. Secondly, it was not solely focused on studies evaluating policy changes. The review also included studies that investigated variations in incentives that resulted from labour market trends (e.g. changing wages relative to benefit levels) and cross-sectional variation in eligibility between jurisdictions within a country (e.g. Canadian Provinces), without a policy change being implemented. Thirdly, it included studies that estimated policy effects on benefit receipt which may not translate into employment effects.

Examining the effect of changes to disability benefits is particularly important because, if recent policies to restrict access are not leading to increased employment, the result of the restriction may be increased numbers of people out of work with disabilities who are not able to secure sufficient income through social protection systems, putting them at greater risk of poverty [11]. This process would be exacerbated by the increases in retirement age currently taking place in many OECD countries, leading to an increase in the proportion of older people out of work with a disability who are no longer eligible for disability benefit or state pensions.

There is an urgent need to synthesise evidence from across OECD countries for the employment effects of changes to eligibility and income replacement rates of disability benefits. It is also important that we understand how these effects might differ between different country contexts, populations groups (e.g. differences by age and gender) and intervention approaches (e.g. expanding or restricting eligibility). We address the gap in the literature by conducting a systematic review of all studies from OECD countries investigating the employment effect of changes to the eligibility or income replacement rates of disability benefits amongst older working age people.

## Materials and methods

Through our search and selection strategy we sought to identify all empirical studies published between 1990 to 2018 from OECD countries that addressed the research question: 'To what extent do policy changes that influence the eligibility requirements and/or income replacement rates of disability benefit programmes affect employment for disabled people?'. The systematic review protocol was registered with the Prospective Register for Systematic Reviews (PROS-PERO) (Registration code: PROSPERO 2018 CRD42018103930).

### Search strategy

Electronic databases MEDLINE and MEDLINE In Process and other Non-Indexed Citations, EMBASE, PsycInfo, and Econlit (published and working papers) were searched using a broad set of search terms to identify all studies evaluating the effect of macro-level policies (implemented at the country or regional level) on the employment of people with long-term health problems or disabilities. Searches were made according to the PICO (Table 1), and search terms were customized for each database, and included a combination of subject heading and text words. An example of the MEDLINE search strategy is shown in S1 Appendix.

### Selection

The titles and abstracts were initially screened and those mentioning changes in disability benefit policies were initially selected. The titles and abstracts were then reviewed against the inclusion/exclusion criteria in Table 1. We limited included studies to those that examined a disability benefit policy change that included data before and after the policy change and incorporated older workers (aged 50–65, upper age limit represents most common retirement age) [12]. We defined disability benefits as 'state supported income replacement benefits paid to individuals out of the labour market due to health problems or disabilities', excluding temporary sickness benefits, for example those covering sickness periods of less than one month. We defined eligibility requirements as any criteria or procedures the applicant needs to meet or undergo in order to be eligible for disability benefits, including the assessment process for ascertaining the presence and level of disability.

Titles and abstracts were screened independently for inclusion by two reviewers. Full-text copies of all papers included during title and abstract screening were then independently screened by two reviewers. During screening, any queries or disagreements were resolved by

**Table 1. Inclusion/Exclusion criteria for systematic review.**

*Inclusion / exclusion criteria*

| | Include | Exclude |
|---|---|---|
| Population | Older working age population (aged 50–65 years), in OECD countries. | Persons younger than 50 years of age, or older than 65 years of age. |
| | | All other countries. |
| Intervention | Changes in the income replacement level, eligibility and/or assessment approaches of disability benefits and long-term sickness benefits. | Changes to other forms of disability benefits. |
| | | Changes to temporary sickness benefits. |
| | | Changes to other forms of income replacement benefits. |
| | | All other types of benefits. |
| Comparison | Either comparisons with the same population prior to the policy introduction (e.g. as in before and after and interrupted time series studies), or comparison over time between populations experiencing the policy change and those who have not. | Cross sectional studies of those that only included the exposed population. |
| Outcomes | Effect on the probability of being in employment or participating in the labour market. | Volunteer work. |
| | | Length of time on disability/sickness benefits. |
| Study designs | Studies that include data pre and post policy exposures including: | Studies that do not include data pre and post policy exposures. |
| | Controlled intervention studies | |
| | Before and after studies Interrupted time series | |
| | studies | |
| | Difference in differences Panel regression studies. | |

*Publication characteristics inclusion / exclusion criteria*

| | Include | Exclude |
|---|---|---|
| Publication types | Primary empirical studies from peer-reviewed literature. | Any work that is not a primary empirical study, including editorials, opinion and discussion pieces. |
| | Papers published or in-press. | |
| | Working papers. | Previous reviews and meta-analyses. Relevant reviews were, however, used to identify relevant primary studies. |
| Year of publication | 1990–2018 | Prior to 1990 |
| Language | English language | Non-English language |

discussion, or by recourse to a third reviewer. Reference lists of all included studies were hand-searched to identify further studies of interest (i.e. 'backward citation searching'). Forward citation searches of included studies were also conducted using Web of Science. The screening process was conducted using EPPI-Reviewer 4 systematic review management software [13].

## Data extraction, methodological quality assessment, and evidence synthesis

Data from each included study were extracted into pre-designed and piloted forms. Forms were completed by one reviewer and checked for accuracy by another. All data extracted from included studies was double assessed. Data extracted included study design, population, sample size, year and duration of study, intervention type, outcomes, controls, results. Extracted data was collated in a structured database.

Methodological quality assessment (QAs) was independently conducted by two reviewers, using a framework adapted from Barr et al (2010) [10]. Any disagreements were resolved by

discussion, or by recourse to a third reviewer. S2 Appendix shows the domains and method of quality scoring. The QA was designed to distinguish strengths of the methodological approaches for natural experiments, and the maximum score was 21.

Studies were also narratively synthesised [14], with higher quality studies being reported first and in greater detail [15]. We synthesised separate results from each paper with each paper potentially contributing multiple results–from evaluations of multiple reforms, from analysis of single reforms on multiple population groups (e.g. men and women) or from analysis of single reforms using different datasets. For papers that reported results from multiple regression models (e.g. controlling for different covariates) we selected the results which had been adjusted for the highest number of appropriate covariates. Harvest plots were initially used to display and summarise the results of the included studies, and to explore variation within subgroups of study based on type of policy intervention, employment effect, quality and country [16]. A Harvest plot is a matrix which allows results to be plotted by subgroups and displayed in a way which incorporates all relevant results.

## Meta-analysis

To visualise variation in study effect sizes results and perform some exploratory meta-analysis where possible, all results were transformed to provide the relative risks (RR) of being in employment after the policy change was implemented. Where the policy effect was presented as an absolute measures of the percentage point change in employment, this was converted to a relative measure by using the baseline employment rate at the denominator. Where the baseline employment rate was not available in the paper, this was taken from national statistics data for the relevant population. Odds ratios were converted to RR using the method described by Zhang and Yu [17].

As noted above the extent to which policy effects vary between different country contexts, populations groups (e.g. differences by age and gender) and intervention approaches (e.g. expanding or restricting eligibility), is not known. We therefore investigate these differences using random effects meta-regression. Firstly, we investigate whether the effects of restricted eligibility are similar to the inverse effects of expanded eligibility. We applied a reciprocal transformation to the relative risks (RR) of studies of reforms which expanded eligibility, while RRs from studies of reforms which restricted eligibility were unchanged. Therefore, if a study found that a policy expanding eligibility decreased employment by 10% (RR = 0.9) in the meta-regression this transformed to an RR of 1/0.9 = 1.11. We then used meta-regression to test whether the effect sizes were different between these two study types i.e. whether it was reasonable to assume that the effects of restricting eligibility are simply the reciprocal of the effects of expanding eligibility. We then used meta-regression to explore heterogeneity of effects by several stratifications; USA and other countries, sex specific, limited to older age populations ($> = 50$ years old), decade of policy reform (1980s, 1990s 2000s), Quality Appraisal score (above or below the median score of 17). In a final meta-regression we estimated the pooled effect size across all studies. The transformed RRs were additionally plotted on a forest plot to visualise the effect size and on a funnel plot with their standard errors to investigate the level of publication bias, which was additionally assessed with the Egger's test [18].

## Results

Our initial searches identified 19,670 potentially relevant studies after deduplication, and 18 publications were eventually identified that met our inclusion criteria [9,19–35]. Fig 1 shows the progression of studies through the review process. The 18 studies included were from

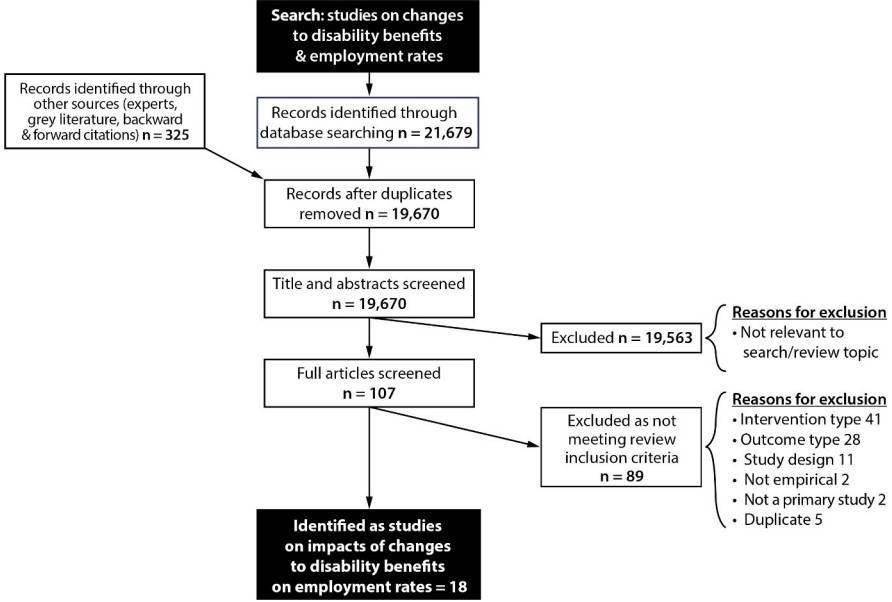

**Fig 1. PRISMA flow chart.**

seven countries in total; 4 from USA, 2 from UK, 7 from Canada, 2 from the Netherlands, and one each from Spain, Sweden, and Austria.

Eight studies evaluated a policy change which expanded eligibility criteria for disability benefit, nine evaluated a restriction of eligibility criteria and only one study investigated a change in income replacement levels without changes in eligibility. As there was only one study that investigated a change in income replacement levels alone [35], we excluded this study from further analysis and focused our synthesis on the 17 studies investigating changes in eligibility.

## Policies evaluated

Table 2 shows the details of these 17 studies and the policies they evaluated. The policy reforms were introduced between 1984 and 2010, with the majority of reforms occurring in the 1990s. Most of the reforms included a change in the assessment process for disability benefits (15/17) [9,21–30,32,33]. This included changes to: the criteria used to assess a claimants level of impairment; the set of jobs disability was assessed against–typically changing from a claimants usual job to any job (or vice versa); whether to account for wider social circumstances; and changes to the person carrying out the assessment–typically changed from the claimants family doctor to a government appointed independent assessor. Whilst most of the studies (14/17) investigated reforms of national disability benefit schemes, three US studies investigated a specific change to the Veterans' Affairs Disability Compensation program that extended the assessment criteria to include diabetes after it had been shown that exposure to Agent Orange increased risk of diabetes. As this scheme is only available to US Veterans the findings of these studies may have limited generalisability to broader populations.

Some of studies also evaluated policy reforms that included changes to eligibility requirements related to prior earnings requirements (5/17). With contributory schemes claimants may only be eligible for benefits if they have worked during a certain proportion of years prior to claiming. A number of reforms have changed these rules–potentially increasing or decreasing the numbers of people eligible. Finally, the reforms evaluated in four studies included a change in the benefit income replacement rate as well as changes to eligibility. The difference-

**Table 2. Characteristics of included studies.**

| | Authors | Country | Age | Sex | Policy year | Policy | Subpopulation | Effect on employment | QA |
|---|---|---|---|---|---|---|---|---|---|
| **Expanding eligibility** | | | | | | | | | |
| 1 | Autor and Duggan, 2003 [9] | USA | 25–54 | M&F | 1984 | Change to the federal Disability Insurance (DI) program that introduced a broader definition of disability providing applicants and medical providers with greater opportunity to influence the decision process. | Male, low education | Decrease | 18 |
| | | | | | | | Male, High education | NS | |
| | | | | | | | Female, low education | Decrease | |
| | | | | | | | Female, high education | NS | |
| 2 | Gruber 2000 [31] | Canada | 45–59 | M | 1987 | 1987- Changes in Canada Pension Plan (CPP) disability programme that included reducing the required earnings history be eligible, increasing flat rate component by 150% representing a rise of 36% in the replacement rate relative to the Quebec programme and introduction of an early retirement option at age 60. | NA | Decrease | 18 |
| 3 | Campolieti, 2003 [23] | Canada | 45–64 | M | 1989 | Change in Canada Pension Plan (CPP) disability program eligibility criteria to incorporate socioeconomic conditions (e.g. high regional unemployment, a person's skills and the lack of particular sorts of jobs in a region) to qualify for CPP disability benefits—in contrast to solely using medical criteria. | NA | Decrease | 17 |
| 4 | Autor and Duggan, 2007 [19] | USA | 49–60 | M | 2001 | 2001 change to eligibility criteria of Veterans' Affairs Disability Compensation program (VDC)—in which to be eligible, a veteran's disability must be caused or aggravated by military service. In 2001 the criteria were extended to include diabetes for veterans who served in Vietnam War (after evidence that Agent Orange herbicide exposure was linked to diabetes). | NA | Decrease | 17 |
| 5 | Autor et al., 2015 [20] | USA | 43–65 | M | 2001 | Same widening of eligibility criteria of Veterans' Affairs Disability Compensation program (VDC) as outlined for Autor and Duggan 2007 [19] | NA | Decrease | 17 |
| 6 | Duggan 2006 [28] | USA | 47–63 | M | 2001 | Same widening of eligibility criteria of Veterans' Affairs Disability Compensation program (VDC) as outlined for Autor and Duggan 2007 [19] | NA | NS | 14 |
| 7 | Campolieti, 2001 [24] | Canada | 45–64 | M&F | 1987–1989 and 1993 | (1) Period of relaxation of eligibility requirements for the CPP 1987–1989 as described for 1987 by Gruber (2000) [31] and 1989 for Campolieti (2003) [23]. (2) 1993 QPP relaxed eligibility in QPP -changed requirement for being unable to work from "any job" to "usual job" and relaxed the contribution requirements. | Female QPP | Increase | 14 |
| | | | | | | | Female CPP | NS | |
| | | | | | | | Male QPP | NS | |
| | | | | | | | Male CPP | NS | |
| 8 | Campolieti, 2001a [25] | Canada | 45–64 | M | 1987–1989 and 1993 | The same policies relaxing eligibility in CPP in 1987 and 1989 as outlined for Gruber 2000 [31] and Campolieti 2003 [23]. | NA | NS | 12 |
| **Restricting eligibility** | | | | | | | | | |
| 9 | Disney et al., 2003 [29] | UK | 57–71 | M&F | 1995 | Reform to disability benefits introducing stricter assessment process carried out by a government-approved doctor as opposed to the claimants' family doctor. Assessment based on the claimant's ability to carry out any work as opposed to previously whether they could undertake their usual work. The benefit was limited to those under the state pension age and the earnings-related component was removed reducing the income replacement rate considerably for some groups. | NA | NS | 19 |

*(Continued)*

**Table 2.** (Continued)

| | Authors | Country | Age | Sex | Policy year | Policy | Subpopulation | Effect on employment | QA |
|---|---|---|---|---|---|---|---|---|---|
| 10 | Barr et al., 2016 [21] | UK | 18–64 | M&F | 2010 | The introduction of stricter assessment process—the Work Capability Assessment (WCA), a functional-abilities checklist that uses a point-based system to determine eligibility. In 2010 this new assessment was applied to all 1.5 million existing disability benefit claimants. | Male, mental health problem | NS | 18 |
| | | | | | | | Male, physical health problem | Decrease | |
| | | | | | | | Female, mental health problem | Decrease | |
| | | | | | | | Female, physical health problem | NS | |
| | | | | | | | M&F, mental health problem | NS | |
| | | | | | | | M&F, physical health problem | NS | |
| 11 | Borghans et al., 2012 [22] | Netherlands | 42–59 | M&F | 1993 | 1993 reform outlined above for de Vos 2011 [27], introducing stricter disability assessment criteria, reassessment and time limited benefits for younger claimants. | NA | Increase | 18 |
| 12 | Staubli 2011 [33] | Austria | 55–56 | M | 1996 | 1996 reform—tightening of eligibility criteria for men aged 55–57 who were previously eligible if their ability to work in a similar occupation was reduced, from 1996 they were assessed on their ability to work in any occupation. | NA | Increase | 18 |
| 13 | Tanaka, Hsuan-Chih and Nguyen, 2016 [34] | Canada | 16–64 | M&F | 1997 | 1997—Change in Canadian Pension Plan (CPP) disability programme that increased the number of years of prior employment required for eligibility. | Female, 46–50 | NS | 18 |
| | | | | | | | Female, 51–55 | NS | |
| | | | | | | | Female, 56–60 | NS | |
| | | | | | | | Male, 46–50 | NS | |
| | | | | | | | Male, 51–55 | NS | |
| | | | | | | | Male, 56–60 | NS | |
| 14 | Campolieti and Goldenberg, 2007 [26] | Canada | 45–64 | M&F | 1995–1996 | Changes to the CPP disability programme that introduced more stringent medical criteria and removed the incorporation of social and economic factors in decision making. This was combined with the decentralisation of disability assessments to regional offices. | Female, NPHS | NS | 17 |
| | | | | | | | Female, SCF | NS | |
| | | | | | | | Male, NPHS | Decrease | |
| | | | | | | | Male, SCF | NS | |
| 15 | Karlström, Palme, and Svensson, 2008 [32] | Sweden | 60–64 | M | 1997 | Introduction of stricter assessment criteria for older workers including stricter medical requirements, judged in relation to all jobs not just previous occupation or jobs in local area and requirement to engage in rehabilitation. | NA | NS | 16 |
| 16 | de Vos et al., 2011 [27] | Netherlands | 50–63 | M&F | 1993–2006 | Multiple reforms from 1993–2006. Including: 1993—Introducing stricter disability assessment criteria, reassessment and time limited benefits for younger claimants 1996—Requirement for employers to pay 70% of earnings for 1 year and increased prior earnings requirements for eligibility. 2002—improved gatekeeper role and requirements for reintegration into employment. 2004—stricter re-assessment requirements for younger claimants. 2006—Introduction of strict distinction between partially and fully disabled. | 2006 reform | NS | 16 |
| | | | | | | | 2002 reform | NS | |
| | | | | | | | 1993 reform | NS | |
| | | | | | | | 1996 reform | Increase | |
| | | | | | | | 2004 reform | NS | |
| 17 | Garcia-Gomez, Jimenez-Martin and Castello, 2011 [30] | Spain | 50–64 | M&F | 1997 | Introduction of stricter assessment of disability, replacing assessment against current job to usual occupation, new independent assessment team replacing assessment by claimants' own doctor. | NA | NS | 15 |

QA- Quality Assessment score, CPP- Canada Pension Plan, QPP- Quebec Pension Plan, SCF- Survey of Consumer Finances, NPHS- National Population Health Survey

in-difference approach was the most common design strategy (12/17), four used an interrupted time series approach, and one used an instrumental variable estimate.

## Narrative synthesis of policy effects

From these 17 studies we included 40 separate results (selecting regression estimates that were most adjusted/most appropriate for our research question). Fig 2 presents a harvest plot of these results, showing whether the results indicated the policy was associated with a significant reduction or increase in employment at the 5% level, or if the study found no significant

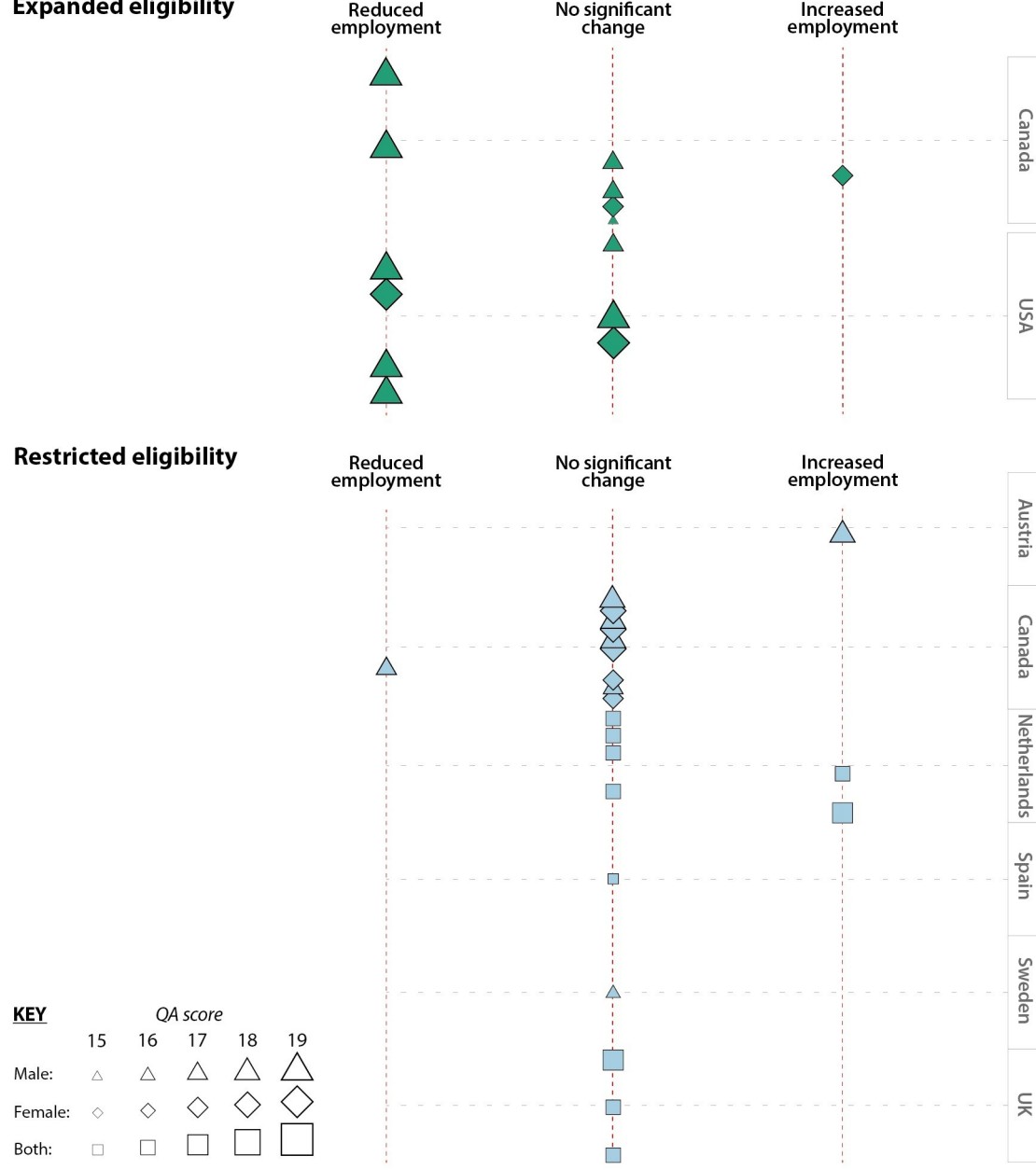

**Fig 2. Harvest plot for employment outcome after reform, stratified by reform type (expansion or restriction), country, sex and QA score.**

difference. Of the six results from Barr et al (2016), we include the two results for the total population (by mental or physical health condition), and exclude the four sex specific results to avoid repetition of findings from the same study and population, as this would give undue weight to this study in the visualisation and meta-regression [21].

Fourteen results were related to the effect of expanded eligibility criteria, from two countries (USA and Canada). Six of these results showed that expanded eligibility was significantly associated with reduced employment (all from above average quality studies, four from USA). Two of these studies found that the expansion of the US Veterans' Affairs Disability Compensation program (VDC) to include people with diabetes was associated with a decline in employment amongst male veterans. Autor et al. (2015) found that 18 percent of all newly eligible veterans left the labour force, while Autor and Duggan (2007) concluded that the policy change was associated with a significant, and substantial, reduction in employment among veterans [19,20]. Autor and Duggan (2003) also found that the use of a broader definition of disability in the US Disability benefit programme from 1984 was associated with reduced employment of low educated older men and women, but did not have the same effect on higher educated groups [9].

Between 1987 and 1994, the eligibility for the Canadian Pension Plan (CPP) was relaxed as a result of policy reforms in 1987 and 1989. The 1987 reform reduced the prior earning requirements and substantially increased the income replacement rate of the benefit and the 1989 reform incorporated socioeconomic conditions in the assessment process. Campolieti (2003) found that the 1989 was associated with a significant reduction in male employment [23]. Gruber (2000) found the 1987 reform was associated with a significant reduction in male employment, however this study did not separate the effect of changing eligibility criteria from the increased replacement rate [31]. In contrast, earlier analyses by Campolieti (2001, 2001A) found that the policy changes which relaxed CPP eligibility criteria between 1987 and 1994 had no significant effect on employment, and separated the effects of changed replacement rate from eligibility criteria [24,25]. Campolieti (2001) also studied a relaxation of the assessment requirements in the Quebec Pension Plan (QPP) disability program from being unable to do any job to being unable to do their "usual job". This study found no significant association between the QPP reform and male employment, but was associated with an increase in female employment, the opposite effect to that which was expected [25].

Nine studies investigated the impacts of restricting eligibility criteria giving rise to 22 results. Three of these results indicated that restricted eligibility was significantly associated with increased employment (all above average quality studies, two from the Netherlands, one from Austria). Staubli (2011) found that a 1996 policy in Austria introducing stricter assessment criteria was associated with an increase in employment. This policy changed the assessment process in Austria from judging disability based on ability to do "usual" job to ability to do "any" job. Staubli also noted variation based on education and earnings, and spill over into other welfare programmes. The effect on employment was greater in 'blue-collar' compared with 'white-collar' workers [33].

de Vos and colleagues analysed the effect of 5 reforms between 1993 and 2006 in the Netherlands that involved some element of restricting eligibility to disability and sickness benefits. They found that the 1996 reform was the only one associated with an increase in employment [27]. Whilst this reform did involve some changes in eligibility criteria related to prior earnings, the major change was the requirement for employers to pay 70% of earnings for the first year of sickness or disability. It is likely that the employment effect associated with this reform reflects greater retention in employment due to the increase in employer incentives, than a change in eligibility criteria. Although de Vos found that there was no significant effect of the 1993 reform in the Netherlands that introduced stricter disability assessment criteria, a higher

quality study by Borghans et al (2012) concluded this reform had led to an increase in employment [22].

Eighteen results showed no significant association between policies restricting eligibility and employment (twelve from above average quality studies). Disney et al. (2003) concluded that there were no significant employment effects of a 1995 UK policy change that introduced a new assessment process carried out by a government-approved doctor assessing eligibility based on the claimant's ability to carry out any work as opposed to previously whether they could undertake their usual work [29]. Barr and colleagues concluded that a UK policy reassessing existing claimants of disability benefits claimants under stricter assessment process–the Work Capability Assessment in 2011, did not increase employment [21]. Rather they found it increased transitions of people with mental health problems from disability benefits into unemployment benefits.

In Canada Campolieti and Goldenburg (2007) found that 1995–1996 changes to the QPP disability that introduced more stringent medical criteria and removed the incorporation of social and economic factors in decision making had no significant effect on employment [26]. Tanaka et al (2016) found that a change in CPP disability programme that increased the number of years of prior employment required for eligibility had no significant impact on chances of employment for disabled people [34].

Karlstrom and colleagues concluded that changes in Sweden in 1997 introducing stricter assessment criteria (assessed against any job rather than previous job and stricter medical requirements) did not lead to increased employment [32]. Similarly Garcia-Gomez and colleagues found that a policy replacing assessment against current job to usual occupation and introducing a new independent assessment team in Spain no significant effect on employment [30].

Of the 36 results included, ten examined both sexes combined, 17 focussed on male employment, and 9 focussed on female employment. Methodological Quality Assessment (QA) scores ranged from 12 to 19, out of a potential 21, with an average score of 16.6. Overall, the QA scores suggested that the methodological quality of included studies was high, with no studies scoring below half in our QA score. Eleven of the 17 studies were above average. The majority of findings from these studies show that the policy reforms had no significant association with employment.

## Exploratory meta-analysis

Relative risks of the policy effects were calculated for 39 of the 40 results. The relative risk could not be calculated for one study as the results were presented per 1 percentage point increase benefit enrolment, rather than as the overall policy effect [20]. Only two studies investigated effects by socioeconomic group and therefore we could not analyse differences in the meta-analysis. One of these studies found that expanding eligibility reduced the employment of low educated men and women, and the other found that increased employment after restricting eligibility was larger among 'blue collar' and lower income workers. Table 3 shows the pooled effect size from a random effects meta-regression across different subgroups of studies and populations. As outlined above, for all analysis except the sex specific analysis we excluded the sex-specific results from Barr et al as the results of the entire population were available, leaving 35 results in total [21]. There was a large amount of unexplained heterogeneity between the studies as indicated by the $I^2$ statistics, therefore any meta-analysis should be treated with caution. We present the meta-analysis as a means of exploring the factors that may underly the heterogeneity of the results.

Firstly, we found no evidence that the effect of restricting eligibility criteria on employment was any different than the inverse of the effect of expanding eligibility criteria. This suggests

**Table 3. Pooled risk ratio estimates, from meta-regressions, showing the relative increase in employment associated with policy implementation for sub-groups of studies and overall effect.**

| Inclusion | N | RR | LCL | UCL | p-value for point estimate | p-value for difference in effect size between strata | $I^2$ statistic |
|---|---|---|---|---|---|---|---|
| **Policy type** | | | | | | | |
| Expanding eligibility | 13 | 1.006 | 0.995 | 1.017 | 0.255 | 0.963 | 77.2 |
| Restricting eligibility | 22 | 1.007 | 0.995 | 1.018 | 0.261 | | |
| **Study Quality** | | | | | | | |
| QA score >17 | 14 | 1.012 | 1.001 | 1.024 | 0.036 | 0.163 | 75.5 |
| QA score <17 | 21 | 1.002 | 0.991 | 1.012 | 0.760 | | |
| **Country** | | | | | | | |
| Not USA | 29 | 1.006 | 0.996 | 1.016 | 0.231 | 0.859 | 77.3 |
| USA only | 6 | 1.007 | 0.994 | 1.021 | 0.283 | | |
| **Sex** | | | | | | | |
| Women | 11 | 1.001 | 0.987 | 1.015 | 0.889 | 0.110 | 76.8 |
| Men | 18 | 1.007 | 0.996 | 1.017 | 0.210 | 0.174 | |
| Both sexes | 10 | 1.032 | 0.997 | 1.069 | 0.076 | | |
| **Age** | | | | | | | |
| Only included > = 50–65 year olds | 13 | 1.016 | 0.997 | 1.035 | 0.095 | 0.258 | 68.6 |
| Also included <50 year olds | 22 | 1.004 | 0.996 | 1.012 | 0.288 | | |
| **Decade of policy intervention** | | | | | | | |
| 1980–1989 | 9 | 1.006 | 0.993 | 1.019 | 0.342 | | 77.6 |
| 1990–1999 | 19 | 1.005 | 0.994 | 1.016 | 0.390 | 0.863 | |
| Post 2000 | 7 | 1.015 | 0.990 | 1.040 | 0.230 | 0.538 | |
| **All** | **35** | **1.006** | **0.999** | **1.014** | **0.107** | | **76.6** |

RR- relative risk of employment, LCL- Lower confidence interval, UCL–Upper confidence Interval

that it is a reasonable assumption to transform the effect sizes of these studies in the further meta- analysis. Limiting the metanalysis to those studies assessed as being of higher than average quality did indicate a larger, significant effect size (p = 0.036), compared to lower quality studies. There was no evidence indicating a difference in effect between studies of US policies compared to other countries. Studies investigating the policy impact on both men and women tended to show larger effect sizes than results of men and women separately, in particular the 11 studies presenting results for specifically for women tended to show very small effects, although none of these differences by gender were statistically significant. There was some evidence that studies just focusing on older workers indicated slightly larger effect sizes with the point estimate of borderline significance in this group (p = 0.095). Although studies of more recent policies in the 2000s indicate larger effects these differences were not statistically significant.

Pooling effect sizes across all studies indicates an overall effect size 1.006 (95% CI 0.999–1.014). Whilst concerns remain about pooling across such heterogenous studies the lack of a difference between sub-groups in Table 3, suggests this may not be a problem.

Fig 3 shows forest plots for RR and confidence intervals for each of the results, indicating these largely cluster around an RR of 1 with the more precise results tending to be closer to 1, indicating no effect or a small effect.

Fig 4 shows a funnel plot the RR for each study against their standard errors. The effect estimates centred around a RR of 1, with the spread of effect sizes increasing symmetrically as the standard error increases. This symmetric pattern suggests that there was no evidence of publication bias in the identification of studies, and this is further supported statistically; the Egger's test was not significant (p = 0.132).

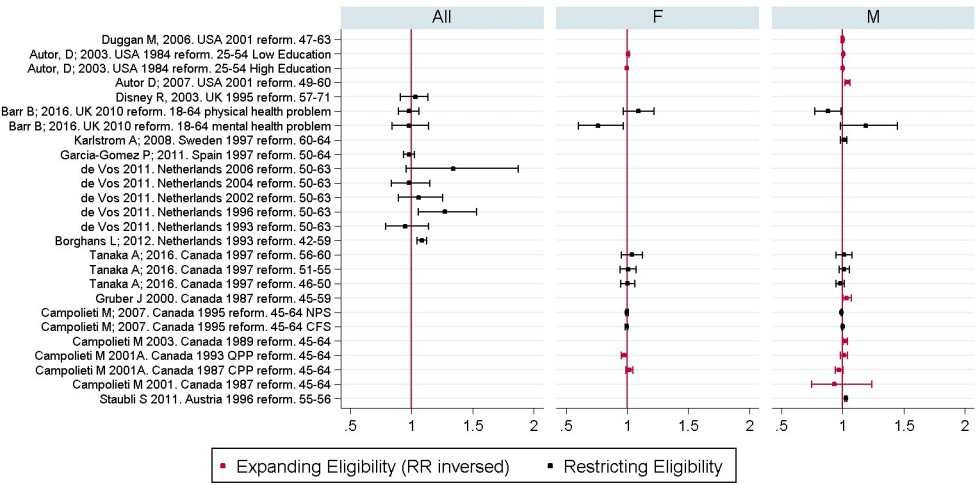

**Fig 3. Forest plot showing the relative effect of restricting eligibility to disability benefits on employment, by age group by sex.** Note: Colum labels show, author, publication year, year of policy reform, age group. For Barr (2016) [21] separate results are given for people with mental and physical health problems, for Autor (2003) [9] separate results are given for high and low educated groups, for Campolieti (2007) [26] separate results are also given for analysis using National Population Health Survey (NPS) and Survey of Consumer Finances (CFS).

## Discussion

Our systematic review aimed to determine whether changing eligibility criteria for disability benefit influenced the employment of disabled people in OECD countries and whether this varied between countries, populations groups and intervention approaches. Overall, we found no consistent evidence that either expanding eligibility reduced employment or that restricting eligibility increased employment. Although there was some evidence of an effect in the hypothesised direction when limiting the analysis to higher quality studies, there is a high level

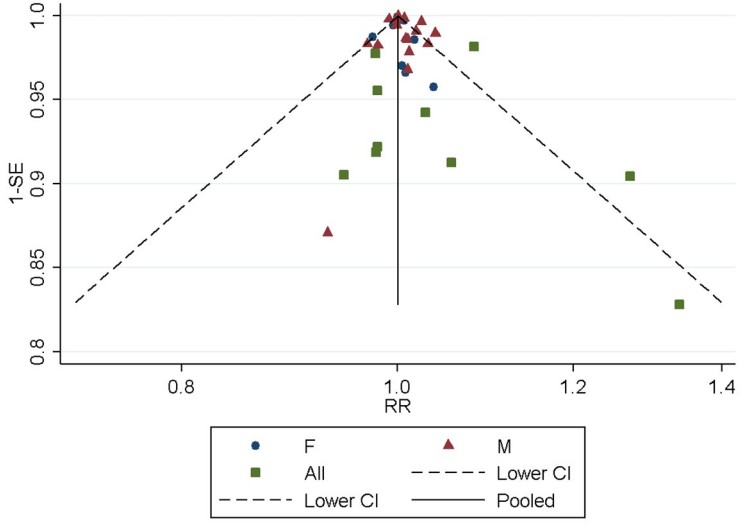

**Fig 4. Funnel plot for included studies, RR against 1 –standard error.**

of uncertainty about this finding. There was no evidence that effects differed between studies in the US compared to other countries and some weak indication that policy effects may be greater in men, older workers and in more recent decades, although none of these differences were statistically significant at the 5% level. Most of the included studies found no significant association with employment.

The effect size from the pooled analysis reflects the effect of the policy on overall employment. It is important to note that a small effect at the population level could reflect a larger impact on the employment of people with disabilities. For example, applying the pooled effect size [RR 1.006, 95% CI 0.999 to 1.014] estimates to people with disabilities specifically, using the average employment rate and disability prevalence across the seven study countries (see S3 Appendix), would indicate an effect size equivalent to an absolute 2.6 percentage point [95% CI -0.44 to 6.2] increase in the employment rate of people with disabilities. The effect size when limiting the analysis to higher quality studies would indicate a 5.3 percentage point increase is the employment of people with disabilities [95% CI 0.44 to 7.4] (see S3 Appendix for details of calculations). If this was the true effect, it would indicate a moderate impact, however the wide confidence intervals indicate a great deal of uncertainty around this estimate. This also assumes that there are no spill-over effects from these changes on the wider working population.

There are three main potential explanations that we find a lack of consistent evidence of a policy effect. First, it is possible that the included studies are insufficiently powered to detect an effect. Underpowered studies are common in the economics literature [36]. This problem is potentially exacerbated in many of the studies included as they aimed to estimate the effect on overall employment levels and it is possible that the effects on people with disabilities are somewhat hidden by a lack of effect in the wider population. Whilst our systematic review and meta-analysis help overcome some issues with a lack of power, the number of studies and high level of heterogeneity between studies could mean that this was not sufficient to provide precise estimates of effect.

Secondly, it is possible that there was an effect of the policy but because of biases in the study designs they were not able identify this effect. This is an issue with all observational policy evaluations, unobserved confounding factors may explain the observed results rather than the causal impact of the policy. Studies may fail to correctly identify a policy effect because the study design is unable to account for other factors that influence employment that occur at the same time as the policy. Several studies have shown that labour market conditions determine the inflow into disability benefits [9,37], so not taking into account these factors could bias results. The most common study design was difference-in-differences, which should overcome some of these biases, if labour market conditions remained similar between comparison groups, but these studies will still be biased where there are imbalances in factors associated with the trajectories of outcomes between the intervention and comparison groups. Our finding that effect sizes increased when limiting the analysis to high quality studies does suggests that the quality of underlying studies may be biasing estimates towards the null.

Thirdly, changing eligibility criteria for disability benefit may have no effect or only a small effect on employment outcomes for disabled people. This could be because prevailing labour market conditions for disabled people at the times when these policies were implemented dampened any potential policy effect. It has been suggested that the timing of policy implementation in the economic cycle may influence effectiveness [38]. Evidence shows that during economic downturns, demand for workers with disabilities declines, and a main driver of employment outcomes in the disabled population is due to demography and initial employment status [37,39]. However when we stratified the analysis by decade we did not find strong evidence indicating that this modified the policy effect. The effectiveness of these policies may also vary based on the composition of the population out of work with a disability; for example

due to historic disability policies different countries may have differing fractions of people out of work with a mental health related disabilities which may influence the effectiveness of interventions focused on work incentives [21].

A lack of effect may be because the reforms studied were not substantial enough to have an influence on employment outcomes. Many of the reforms in this review, however, have had a substantial impact on receipt of disability benefits. In this case, whilst restricting access to disability benefits may lead to a decline in benefit receipt these policies may lead claimants to move onto other, less adequate benefits; for example, unemployment benefits which may explain why there is no significant employment effect [21]. Finally lack of effect could be due to the fact that access to disability benefits is not a major factor influencing the employment of people with disabilities, compared to other complex barriers to employment, such as inaccessible work environments, lack of suitable jobs in local labour markets or lack of skills [40,41].

## Limitations

There are a number of limitations with the review. Firstly the review only located evidence from seven of the 36 countries in the OECD, potentially limiting the generalisability of these findings to the wider OECD. Studies from a wider range of member states are needed to indicate the potential impact of these policies across OECD countries.

Secondly, as is common with other systematic reviews, one limitation of our review is that it only covers research available at the time searches were completed in 2018. It is possible that further research has been published since that date. A full update of searches, screening, review and synthesis was not possible within this study. However, to indicate if substantial research had been published after this date, we performed a rapid, abridged search to identify any newly published literature between June 2018 and July 2020. This search identified one potentially relevant study which investigated the effect of the Netherlands reforms of 2002 and 2004, which had already been investigated by de Vos [27,42]. This study found that restricted eligibility had a small, positive employment effect in older male workers, but not female.

Thirdly, the evidence included in this review came from observational studies. Inherent issues with this study design, particularly concerning unobserved confounding, limit the causal inferences we can make. Whilst difference-in-difference design used in many of these studies, allow some time invariant unobserved confounding to be accounted for, results will be biased if unobserved trends in confounders disproportionately affect the intervention group. The instrumental variable approach used in some studies potentially overcomes some of these issues, however, the suitability of instruments in these analyses is often untestable. The use of ecological data in some studies also limits the ability to control for individual level characteristics and introduces the potential for ecological fallacy [24,25,30].

Fourthly, the high level of heterogeneity between studies means that results from the meta-analysis need to be treated with caution. This included significant variation in the demographics of populations studied (sex and age), the policy context in the country, and the characteristics of the policy [43]. It may not be the case that the impact of restricting access is the inverse of expanding access, however in the meta-regression analysis we found no evidence that this was not a reasonable assumption. Only five of the policy reforms investigated included a change in the replacement rate, and in four of these the reform also included a change in eligibility criteria, therefore it was not possible to distinguish between the effects of changing replacement rates and changing eligibility criteria. It was also not possible to investigate whether the policy effect differed based on the amount the replacement rates changed, or whether the effect was modified based on the baseline replacement rate in the different countries. Contrary to our expectation meta-analysis investigating differences in effect between

studies in the US and in other countries did not indicate noticeable differences in effect, similarly we found no strong evidence to indicate differences in effect between age and gender groups. The relatively small number of studies included in the review limits the extent to which these differences could be investigated empirically. The high level of unexplained heterogeneity between studies also highlights that generalisation of the results to other settings and populations should be made with caution.

## Policy implications

Whilst there was a wide variety of policy reforms evaluated by the included studies, the majority of studies (12/17) examined relatively similar changes to the assessment of disability. These changes often included (1) changes in functional / medical assessment criteria, (2) changing who performs the assessment–from the claimants doctor to an independent government appointed official, (3) changing the jobs against which capacity to work was assessed–from a person's usual job to any job (or vice versa).

The results from this review indicate that there is no consistent evidence that the approaches of restricting eligibility to disability benefit increases employment. This finding is important as this is the objective of most recent reforms in OECD countries. The evidence of our review suggests that if there is an employment effect it is likely to be moderate. While restricting eligibility may be effective at reducing the number of people receiving disability benefits, which is also a policy goal in many countries aiming to reduce public expenditure, a lack of an impact on employment has serious potential consequences for people with disabilities [44]. This potentially leads to an increasing number of people who are both out of employment with health problems, and not able to access adequate social protection, potentially increasing their risk of poverty [45]. This increased risk of poverty may lead to further exacerbation of poor health [11,46], further reducing employment prospects and increasing living costs. Extending the age of eligibility for state pensions, may increase this problem, and careful consideration should be made as to the effects of such policy changes poverty outcomes, as well as the employment of older workers.

If reforms to disability benefits lead to increased risk of poverty and subsequent deterioration in health and end up recycling of claimants between benefit schemes, rather than significantly increasing employment, they may also may not even achieve their fiscal goals. In the UK, for example, the introduction of the Work Capability Assessment was expected to save the government £5 billion through reduce claims, however the government's own assessment found that no savings were made, as claimants had largely moved onto other benefits [47]. This presents the possibility that reforms to restrict access to disability benefits lead to "all pain, no gain", disadvantaging people with disabilities whilst not improving their employment prospects. Strategies to reduce the disability employment gap may be more effective if they focused more on policies that have been shown to be effective–return-to-work policies, such as graded return-to-work or support to make adjustments, or other active labour market policies [11,48,49].

Our review has found there is no consistent evidence that changing eligibility criteria has an effect on employment outcomes for disabled people. Given the potential negative effects of these policy reforms, there needs to be consideration of changing focus to other approaches to improve employment outcomes for disabled people.

## Supporting information

**S1 Checklist. PRISMA 2009 checklist.**
(DOC)

**S1 Appendix.**
(DOCX)

**S2 Appendix.**
(DOCX)

**S3 Appendix.**
(DOCX)

## Author Contributions

**Conceptualization:** Ben Barr.

**Data curation:** Philip McHale, Andy Pennington, Quenby Mahood, Ben Barr.

**Formal analysis:** Philip McHale, Ben Barr.

**Investigation:** Philip McHale, Andy Pennington, Ben Barr.

**Methodology:** Philip McHale, Andy Pennington, Ben Barr.

**Validation:** Philip McHale, Andy Pennington.

**Visualization:** Andy Pennington, Ben Barr.

**Writing – original draft:** Philip McHale, Andy Pennington, Ben Barr.

**Writing – review & editing:** Cameron Mustard, Quenby Mahood, Ingelise Andersen, Natasja
    Koitzsch Jensen, Bo Burström, Karsten Thielen, Lisa Harber-Aschan, Ashley McAllister,
    Margaret Whitehead.

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
