## [Decision Letter · Decision Letter 0]

28 Aug 2020

PONE-D-20-19352

What is the effect of changing eligibility criteria for disability benefits on employment? A systematic review and meta-analysis of evidence from OECD countries

PLOS ONE

Dear Dr. McHale,

Thank you for submitting your manuscript to PLOS ONE. After careful consideration, we feel that it has merit but does not fully meet PLOS ONE’s publication criteria as it currently stands. Therefore, we invite you to submit a revised version of the manuscript that addresses the points raised during the review process.

In your revision it is important to address the following: 

1. The comparability of the studies across set of countries (e.g. US vs. non-US), raised by both reviewers R1 and R2,  as well as the comparability over time given that policy interventions occurred under different economic conditions (R2). Presenting evidence separately by group of countries and over time would strengthen this part of the analysis. 

2. Reflect on the assumption of reversibility of policy effects (R1) as well as consider the role of the magnitude of the policy change and the pre-reform differences in benefits generosity (R2). 

3. Clarify several choices such excluding sex-specific results and age restrictions (R1). 

4. Consider the detailed comments by R1 on the consistency of the arguments in several parts of the manuscript. 

5. Consider extending the set of policies as suggested by R2. 

We look forward to receiving your revised manuscript.

Kind regards,

Konstantinos Tatsiramos

Academic Editor

PLOS ONE

2. Please provide additional detail on the methods for the "abridged rapid search" described for recent manuscripts. methods should be described in sufficient detail to enable replication and peer review.

3. Please upload a new copy of Figure 4 as the detail is not clear. Please follow the link for more information: https://blogs.plos.org/plos/2019/06/looking-good-tips-for-creating-your-plos-figures-graphics/

Reviewers' comments:

Reviewer's Responses to Questions

**Comments to the Author**

1. Is the manuscript technically sound, and do the data support the conclusions?

Reviewer #1: Partly

Reviewer #2: Partly

2. Has the statistical analysis been performed appropriately and rigorously? 

Reviewer #1: No

Reviewer #2: Yes

3. Have the authors made all data underlying the findings in their manuscript fully available?

Reviewer #1: Yes

Reviewer #2: Yes

4. Is the manuscript presented in an intelligible fashion and written in standard English?

Reviewer #1: No

Reviewer #2: Yes

5. Review Comments to the Author

Reviewer #1: I have uploaded my full referee report as an attachment. However, the system seems to require that I fill this box with at least 200 characters, despite acknowledging that my report can be uploaded as an attachment.

Reviewer #2: Referee Report for: What is the effect of changing eligibility criteria for disability benefits on employment? A systematic review and meta-analysis of evidence from OECD countries

Manuscript Number: PONE-D-20-19352

See attachment.

6. PLOS authors have the option to publish the peer review history of their article (what does this mean?). If published, this will include your full peer review and any attached files.

Reviewer #1: No

Reviewer #2: No

---

## [Author Response · Author response to Decision Letter 0]

12 Oct 2020

Dear Konstantinos

RE: Response to peer review for PONE-D-20-19352

Thank you for the peer-review reports. We have considered them thoroughly and have given a response to specific points in the reports below. We have also noted there was a coding error for Gruber (2000) when converting the paper result to a relative rate, this has been amended and results have been updated accordingly. With regard to the specific questions you have raised:

1. The comparability of the studies across set of countries (e.g. US vs. non-US), raised by both reviewers R1 and R2, as well as the comparability over time given that policy interventions occurred under different economic conditions (R2). Presenting evidence separately by group of countries and over time would strengthen this part of the analysis. 

We have added table 3 to the results section, which include meta-regressions by separate stratifications of results (by age, sex, USA vs. non-USA and decade of policy reform). We have also expanded the discussion to specifically comment on the issues raised by the comparisons,

2. Reflect on the assumption of reversibility of policy effects (R1) as well as consider the role of the magnitude of the policy change and the pre-reform differences in benefits generosity (R2). 

We have added these issues to the limitations and included a stratification of meta-regression based on whether policy change was an expansion or restriction of disability benefits. Additionally, our narrative analysis already considers these policy approaches separately.

3. Clarify several choices such excluding sex-specific results and age restrictions (R1). 

This has been explained in the manuscript now. See point 11 in response to reviewer 1.

4. Consider the detailed comments by R1 on the consistency of the arguments in several parts of the manuscript. 

See responses to reviewer 1.

5. Consider extending the set of policies as suggested by R2

We have added a short section to the discussion discussing the need to consider other active labour market policies to improve the employment of people with disabilities. We feel a more detailed examination of this is beyond the scope of our paper. We explain in the paper that we were not able to focus on changes in replacement rates in detail because there were very few studies that specifically investigated this and we have added this to the limitations. 

Our specific response to reviewer comments is in the attached letter.

We hope our responses answer the points raised and we welcome any further comments. We have also uploaded updated figure (2, 3 and 4) and a new appendix S3.

Kind Regards

Dr Philip McHale

Clinical Lecturer in Public Health and Policy, University of Liverpool

---

## [Decision Letter · Decision Letter 1]

13 Nov 2020

What is the effect of changing eligibility criteria for disability benefits on employment? A systematic review and meta-analysis of evidence from OECD countries

PONE-D-20-19352R1

Dear Dr. McHale,

We’re pleased to inform you that your manuscript has been judged scientifically suitable for publication and will be formally accepted for publication once it meets all outstanding technical requirements.

Kind regards,

Konstantinos Tatsiramos

Academic Editor

PLOS ONE

Reviewers' comments:

Reviewer's Responses to Questions

**Comments to the Author**

1. If the authors have adequately addressed your comments raised in a previous round of review and you feel that this manuscript is now acceptable for publication, you may indicate that here to bypass the “Comments to the Author” section, enter your conflict of interest statement in the “Confidential to Editor” section, and submit your "Accept" recommendation.

Reviewer #1: All comments have been addressed

Reviewer #2: All comments have been addressed

2. Is the manuscript technically sound, and do the data support the conclusions?

Reviewer #1: Yes

Reviewer #2: Yes

3. Has the statistical analysis been performed appropriately and rigorously? 

Reviewer #1: Yes

Reviewer #2: Yes

4. Have the authors made all data underlying the findings in their manuscript fully available?

Reviewer #1: Yes

Reviewer #2: Yes

5. Is the manuscript presented in an intelligible fashion and written in standard English?

Reviewer #1: Yes

Reviewer #2: Yes

6. Review Comments to the Author

Reviewer #1: I am pleased with the authors' thoroughness and willingness to address each of my original comments. I am satisfied that they have either addressed my comments directly within the manuscript or, in the places they were unable to, they have added clarification or caveats. I believe these changes have improved and refined the already solid paper's foundation and I have no further requests or recommendations.

Reviewer #2: Thank you for the revised version of the manuscript. Most of my comments have been addressed in this new version. Those comments that have not been addressed because of limitations in the data, have been mentioned in the discussion section. Therefore, I am happy with the current version of the manuscript.

7. PLOS authors have the option to publish the peer review history of their article (what does this mean?). If published, this will include your full peer review and any attached files.

Reviewer #1: No

Reviewer #2: No

---

## [Editor Report · Acceptance letter]

18 Nov 2020

PONE-D-20-19352R1 

What is the effect of changing eligibility criteria for disability benefits on employment? A systematic review and meta-analysis of evidence from OECD countries 

Dear Dr. McHale:

I'm pleased to inform you that your manuscript has been deemed suitable for publication in PLOS ONE. Congratulations! Your manuscript is now with our production department. 

Kind regards, 

on behalf of

Prof. Konstantinos Tatsiramos 

Academic Editor

PLOS ONE